

# Analysis of Spatiotemporal variations in mid-upper tropospheric methane during the Wenchuan Ms8.0 earthquake by three indices

Jing Cui[1] and Xuhui Shen[1]

[1]Key Laboratory of Crustal Dynamics, Institute of Crustal Dynamics, China Earthquake Administration, Beijing 100085, China

**Correspondence:** Jing Cui (jingcui_86@yahoo.com)

**Abstract.** This research studied the spatiotemporal variation in methane in the mid-upper troposphere during the Wenchuan earthquake (12 May, 2008) using AIRS retrieval data and discussed the methane anomaly mechanism. Three indices were proposed and used for analysis. Our results show that the methane concentration increased significantly in 2008, with an average increase of $5.12 * 10^{-8}$, compared to the average increase of $1.18 * 10^{-8}$ in the previous five years. The Alice and Diff indices

can be used to identify methane concentration anomalies. The two indices showed that the methane concentration distribution before and after the earthquake broke the distribution features of the background field. As the earthquake approached, areas of high methane concentration gradually converged towards the west side of the epicenter from both ends of the Longmenshan fault zone. Moreover, a large anomalous area was centered at the epicenter eight days before the earthquake occurred, and a trend of strengthening, weakening and strengthening appeared over time. The Gradient index showed that the vertical direction

obviously increased before the main earthquake, and the value was positive. The gradient value is negative during coseismic or postseismic events. The gradient index reflects the gas emission characteristics to some extent. We also determined that the methane release was connected with the deep crust-mantle stress state, as well as microfracture generation and expansion. However, due to the lack of any technical means to accurately identify the source and content of methane in the atmosphere before the earthquake, an in-depth discussion has not been conducted, and further studies on this issue may be needed.

*Copyright statement.*

## 1 Introduction

The great Wenchuan Ms8.0 earthquake, on May 12, 2008, occurred in the Longmenshan Fault zone in western Sichuan Province, China. Its epicenter was located at coordinates $30.95° N, 103.40° E$. The extent of the earthquake and aftershock affected areas in the northeast along the Longmen Shan fault, a thrust structure along the border of the Tibetan Plateau and the

western Sichuan Basin (Figure 1). This earthquake was one of the worst continental earthquake events to have struck China in recent decades, and it killed more than ten thousand people in several cities along the western Sichuan basin. A surface rupture more than 200 km long formed along the Yingxiu-Beichuan fault. The Guanxian-Jiangyou fault was also ruptured by





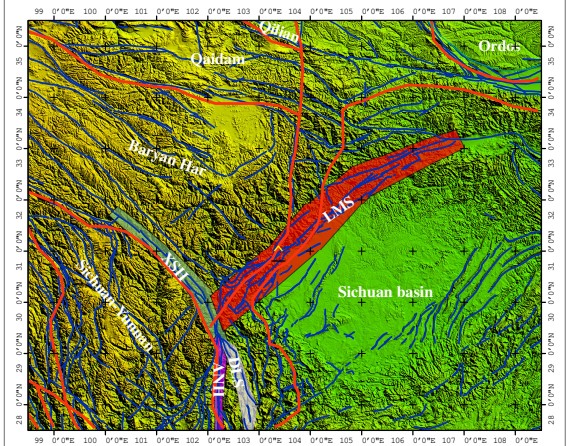

**Figure 1.** Map of the active faults and the location of the Wenchuan earthquake. LMS means the Longmenshan fault zone; XSH means the Xianshuihe fault zone; DLS means the Daliangshan fault zone; Anninghe means the Anninghe fault zone.

surface ruptures that were more than 60 km long, as indicated. The Longmenshan fault zone has a high dipping angle (more than $50° - 60°$) near the surface and a low angle at depth ($15 - 20$ km). This listric shape favors significant strain or energy accumulation, forming large earthquakes. This earthquake was characterized by slow strain accumulation, a long recurrence interval and significant damage power. It is a new type of earthquake that deserves further study (Zhang et al., 2008).

The variation in soil gas concentration serves as a useful tool for monitoring earthquakes. Numerous field investigations have indicated that large amounts of gases are emitted from active fault zones before, during and after great earthquakes. The increased gas emanation from the Earth's crust in the vicinity of active tectonic faults is triggered by a chain of physical processes and chemical reactions from the ground surface. This complex chain leads to geochemical, atmospheric, ionospheric and magnetospheric anomalies (Dobrovolsky et al., 1979; Pulinets and Ouzounov, 2011; Ouzounov et al., 2007). The number of

pre-earthquake thermal, surface latent heat flux and outgoing longwave radiation anomalies apparently results from earthquake-related gas emission from the lithosphere (Tronin et al., 2002; Dey et al., 2004; Tronin, 2006; Ouzounov et al., 2007). Thermal (Zhang et al., 2010), ionospheric (Lin, 2012, 2013; Zhu et al., 2010), electromagnetic (Zhang et al., 2011a), and aerosol (Qin et al., 2014) anomalies were found before the Wenchuan earthquake of May 12, 2008.

As the second most important greenhouse gas after carbon dioxide ($CO_2$), methane ($CH_4$) is approximately 20 times better

at warming the atmosphere than $CO_2$ by weight and plays an important role in atmospheric chemistry. Some people have speculated that thermal abnormalities before earthquakes are related to the release of $CH_4$. The $CH_4$ release mechanism has been investigated and results indicate strong methane emissions when friction is applied to marl-type rock (Martinelli and Plescia, 2005; Italiano et al., 2008). Yue indicated that the Wenchuan earthquake of May 12, 2008, was caused by the rapid migration and expansion of a large amount of highly pressurized and dense $CH_4$ gas in crustal rock masses (Yue, 2013).

Sample results indicated that $CH_4$ was discharged from a shallow reservoir through faults or fractures caused by the earthquake





(Zheng et al., 2013). This study choose to study the spatiotemporal variations of $CH_4$ during the Ms8.0 Wenchuan earthquake from satellite observations.

$CH_4$ satellite observations have started to be applied to seismological studies over recent years. The total column of $CH_4$ associated with the 12 May 2008 Wenchuan earthquake was investigated using satellite data from the AQUA Atmospheric In-
frared Sounder (AIRS), and this work indicates that a large amount of $CH_4$ was emitted from underground into the atmosphere along the Longmenshan Fault Zone from approximately 1.5 months before to 3 months after the earthquake, and the closer to the epicenter, the larger the amount of emitted gas. The peak values were found at intersection areas (Yue, 2013; Cui et al., 2017). However, the same column concentration might correspond to this distinct vertical structure. It is hard to say if the $CH_4$ anomalies are from a ground source or transform. The gas profile can help us understand the terrestrial emission caused by
the earthquake. Recently, NASA declared that the total column values reported for $CH_4$ constituents are dominated by the initial guess of each and should not be used for research purposes (https://docserver.gesdisc.eosdis.nasa.gov/repository/Mission/AIRS/3.7_ScienceDataProductValidation/V6_Data_Disclaimer.pdf). Previous work has shown that an AIRS for tropospheric observation can be used to carry out the study with considerable precision (Zhang et al., 2010; Xiong et al., 2015).

Systematic observation of the vertical variation of $CH_4$ is scarce. The focus of this study is to examine the spatiotemporal
variation of $CH_4$ in the mid-upper troposphere during the Wenchuan earthquake (12 May, 2008) using AIRS retrieval data and to discuss the mechanism of the methane anomaly. Three indices were proposed and used for analysis: the Absolute Local Index of Change of the Environment (ALICE) was used for anomaly detection; the Vertical Concentration Gradient (Gradient) was proposed to study the vertical variation; the Successive Differential Value (Diff) can show the time variation. These three indices analyzed the spatial and temporal distribution of $CH_4$ before and after the earthquake from horizontal, vertical and
time scales helping us understand lithospheric and atmospheric interactions during seismic activity.

## 2 Data and Methods

The Version 6 and Level 3 standard gridded product of 8-day $CH_4$ volume-mixing ratios in a descending model (local night-time), with $1 * 1$ degree of spatial resolution, were obtained from the NASA Goddard Earth Sciences Data and Information Services Center (DISC) (http://disc.gsfc.nasa.gov/AIRS/index.shtml/). The peak sensitivity of the AIRS to methane retrieval
occurs at 300 hPa, and the channels near 7.6 μm are most sensitive to the middle to upper troposphere (Xiong et al., 2015). Therefore, the volume mixing ratio of the middle troposphere (400 hPa, approximately 5 km), the upper middle troposphere (300 hPa, approximately 7 km) and the upper troposphere (200 hPa, approximately 11 km) in the descending orbit data was used.

To extract the methane spatiotemporal anomalies before and after the earthquake, three parameters were applied: the Abso-
lute Local Index of Change of the Environment (ALICE) (Tramutoli, 1998; Tramutoli et al., 2013; Cui et al., 2017) the Vertical Concentration Gradient (Gradient) and the Successive Differential (Diff), all carried out on the basis of eliminating the multiyear background. The background field can partially remove the influence of natural sources such as seasonal changes and surface vegetation, effectively capturing emergency information such as earthquakes, reducing "nonseismic anomalies" to a



certain extent, providing criteria for the extraction of seismic anomalies, and reducing the misjudgment and leakage of seismic anomalies. It was calculated by Eq 1:

$$G_{ref}(x,y,t) = \sum_{i=1}^{N} G_i(x,y,t)/N \tag{1}$$

$$\sigma(x,y,t) = \sqrt{\sum_{i=1}^{N} \left[G_i(x,y,t) - G_{ref}(x,y,t)\right]^2 / (N-1)} \tag{2}$$

where $G_i(x,y,t)$ means the gas value, measured at time $t$, corresponding to a location at (centered on) the coordinates $(x,y)$; $G_{ref}(x,y,t)$ means the reference fields for the study area, defined as a time average gas value; $\sigma(x,y,t)$ is the standard deviation of historical records collected under the temporal constraint. For this study, $N$ was defined as 5 years, from 2003 to 2007. The Absolute Local Index of Change of the Environment (ALICE) is calculated by Eq. 3 (Tramutoli, 1998; Cui et al., 2017):

$$ALICE(x,y,t) = [G(x,y,t) - G_{ref}(x,y,t)/\sigma(x,y,t)] \tag{3}$$

The Vertical Concentration Gradient (Gradient) is used to characterize the vertical variation of gas and is calculated by Eq 4 and Eq 5 :

$$Gradient(x,y,t) = \Delta G(x,y,t,l)/[P_l - P_{l+1}] \tag{4}$$

$$\Delta G(x,y,t,l) = [G(x,y,t,l) - G_{ref}(x,y,t,l)] - [G(x,y,t,l+1) - G_{ref}(x,y,t,l+1)] \tag{5}$$

where $Gradient(x,y,t)$ stands for vertical concentration gradient value, $G(x,y,t,l)$ means the gas value at level $l$. $P_l$ stands for the atmospheric pressure at $l$, $l$ was defined as 3 layers (400 hPa, 300 hPa and 200 hPa). Successive Differential Value (Diff) refers to the difference of the adjacent time gas value and was calculated by Eq. 6:

$$Diff(x,y,t) = \left[G(x,y,t+1) - G_{ref(x,y,t+1)}\right] - \left[G(x,y,t) - G_{ref(x,y,t)}\right] \tag{6}$$

## 3   Results

### 3.1   Reference field

The 8-day average background field at different $CH_4$ heights in the study area was obtained and is shown in Fig.2, 3, and 4. These figures show that the $CH_4$ concentration in the study area decreased significantly from the middle troposphere to the





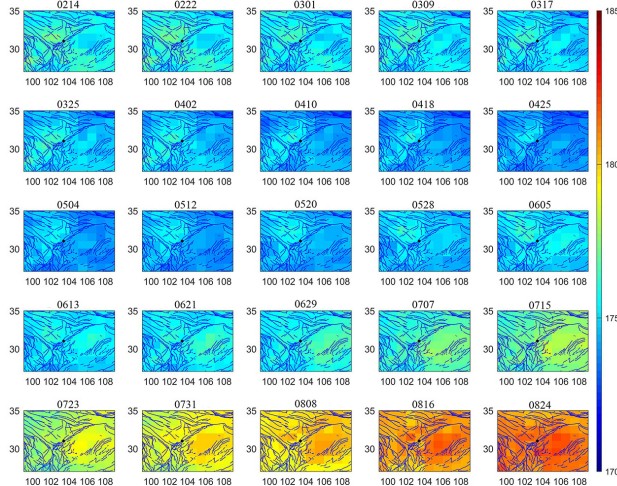

**Figure 2.** Five year (2003 to 2007) average distribution of 8-day nighttime $CH_4$ VMR at 200 hPa.

upper, and the average methane concentrations (from Feb 14th to Aug 31st, 2008) in each stratum were 1.794 PPM, 1.775 PPM and 1.762 PPM for middle, upper middle, and upper troposphere, respectively. This indicates that the middle troposphere is significantly affected by terrestrial emission sources.

The distribution of $CH_4$ has obvious seasonal characteristics and the seasonal cycles at different heights are similar. The $CH_4$ mixing ratio has a weak high value during Feb-Mar, and then decreases during April-May. It starts to increase in June and stays high during July and August, especially over the Sichuan basin. The relatively high values at different times are mostly located at a structural confluence or at tectonic plate boundaries except for late July and August. Earth gas emissions may be the main cause of the high long-term values at that location (http://www.climatechange2013.org/images/report/P36Doc3_WGI-12_Approved-SPM.pdf). Why does the $CH_4$ begin to increase in early summer? Studies have shown $CH_4$ emissions significantly correlate with temperature. When the temperature is $< 15°$ the Earthseldom produces $CH_4$, and when the temperature reaches $35°$, $CH_4$ production can be many times higher than at $25°$ (KazuyukiYagi and KatsuyukiMinami, 1990; Thomas et al., 1996; Holzapfel-Pschorn et al., 1986; Saarnio et al., 2010; Saarnio and Silvola, 1999). Therefore, the $CH_4$ concentration in the troposphere is higher during July and August. The $CH_4$ rice paddy emission, local emissions (such as from gas leakage or energy use), and transport (such as the meridional and zonal advection, convection from the lower troposphere, or stratosphere-troposphere exchange) also increase in early summer (Xiong et al., 2010).

The results of the background field analysis show that the $CH_4$ concentration is greatly influenced by the season, structure and underlying surface, and they present a typical spatial and temporal distribution of methane in southwest China (Zhang et al., 2011b). Gas emanates from the earth's crust continuously, even without earthquakes.



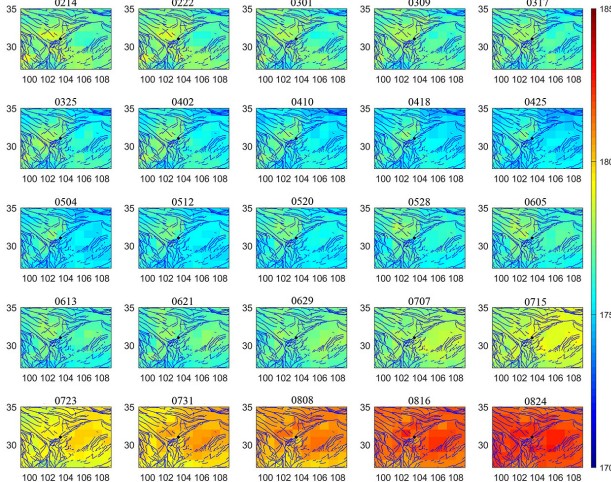

**Figure 3.** Five year (2003 to 2007) average distribution of 8-day nighttime $CH_4$ VMR at 300 hPa

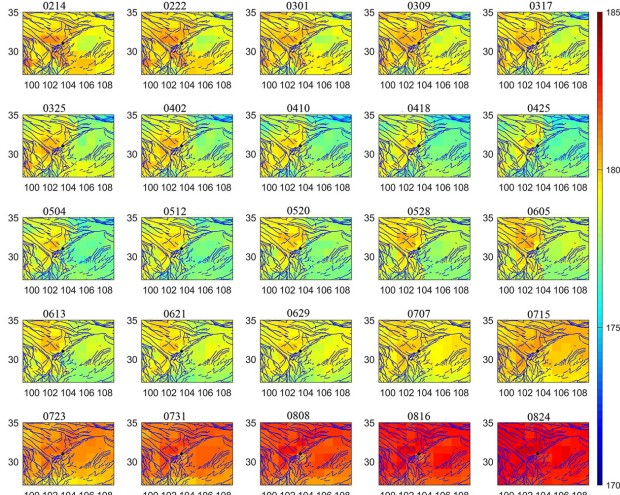

**Figure 4.** Five year (2003 to 2007) average distribution of 8-day nighttime $CH_4$ VMR at 400 hPa

## 3.2 Anomalous Gas

The three indices mentioned above have been applied to identify the spatial and temporal variation of mid-upper tropospheric methane anomalies associated with the Wenchuan Ms 8.0 earthquake using AIRS data.




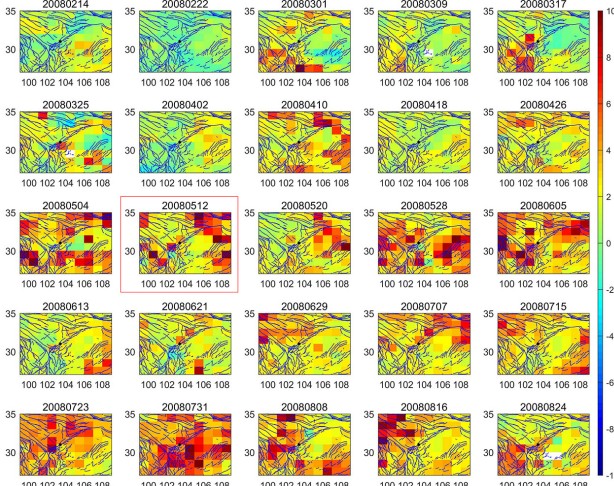

**Figure 5.** Alice index of 8-day $CH_4$ VMR at 200 hPa

### 3.2.1 ALICE index result

Figures 5, 6, and 7 show that as the earthquake approached, the methane concentration near the LMS fault zone gradually increased and the range gradually expanded. High values began to appear in the northern section of the LMS fault zone on March 25th, and began to increase in large areas on May 4th, showing a significant north-south distribution through the LMS fault zone. From May 12th to the 19th, the high value decreased near the epicenter, but remained in large parts of the study area and persisted in large areas after the earthquake. From July 7th, the northern section of the LMS fault zone again showed high accumulation. From July 23rd to August 7th, the high value zone reached the maximum, and all along the LMS fault zone it showed NE distribution. Then, the high-value areas began to weaken on August 8. This circle was probably caused by the big aftershocks on July 24th and August 1st.

### 3.2.2 Vertical Concentration Gradient index result

The Vertical Concentration Gradient stands for the vertical rate of change. Figures 8 and 9 show the value changing obviously at the Bayan har block and plate boundary. The highest values appeared at the Bayan har block, the XSH, ANH and DLS faults and lasted until May 12th before the earthquake. The values were positive before May 12th. They weaken and even turn negative after May 12th, especially during July 23rd to August 7th. The overall shape is still closely related to the fault and basin edge. We will discusses this later.

### 3.2.3 Successive Differential Value

The trend of the reflected gas concentration over time was reflected by the difference change, and the result of 20080222 represented the difference between February 22nd and the 14th. The places with rapid changes mainly occur at plate boundaries





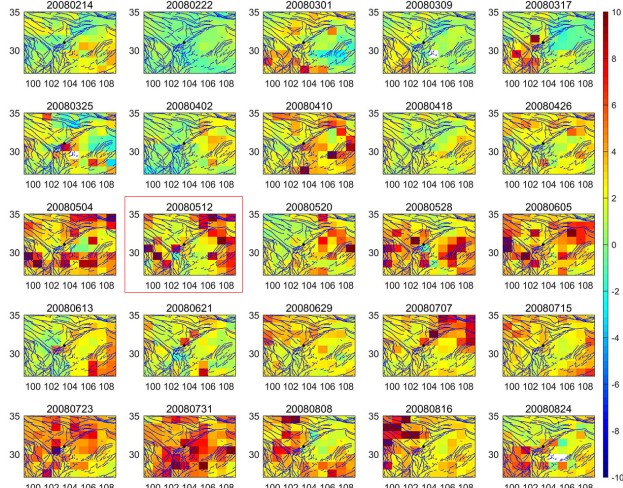

**Figure 6.** Alice index of 8-day $CH_4$ VMR at 300 hPa

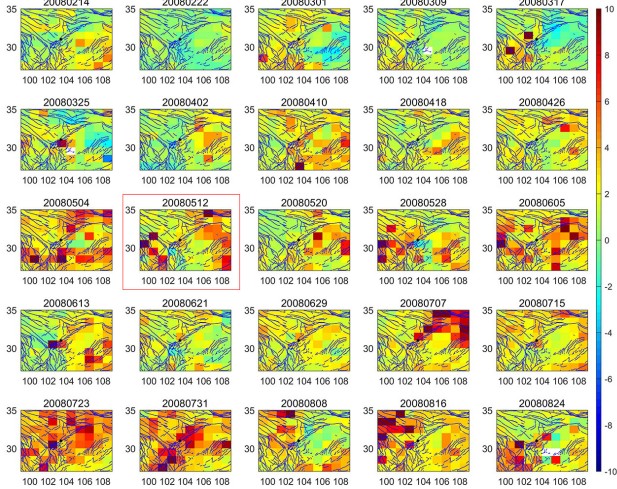

**Figure 7.** Alice index of 8-day $CH_4$ VMR at 400 hPa

and tectonic intersections. Figure 10 shows that the LMS fault zone began to show an abnormally high value on March 25th, and then the high value area gradually increased. On April 10th, high values appeared at the Bayan har block, the XSH, ANH and DLS faults and lasted until May 12th before the earthquake. The value of the entire LMS fault zone was high during May 12th. The next cycle began on July 7th, when high values first appeared in the northern section of the LMS fault zone, followed by the entire LMS, XSH, ANH and DLS fault zones on July 23rd. Two large aftershocks were reported on July 24th and August 1st.





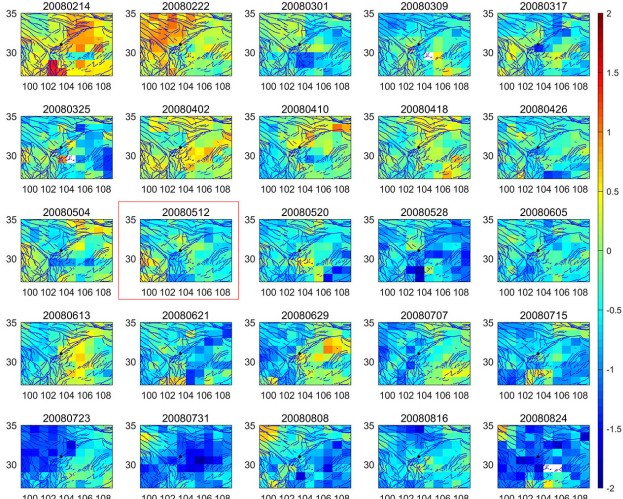

**Figure 8.** Gradient of 400 hPa to 300 hPa

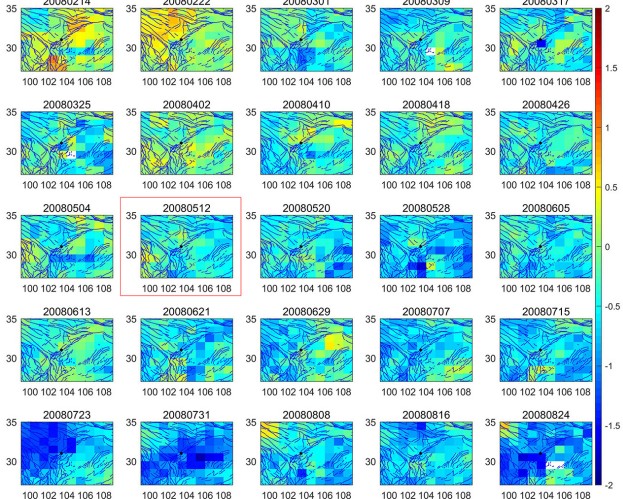

**Figure 9.** Gradient of 300 hPa to 200 hPa

## 4   Discussion

Background field analysis can help us understand the gas distribution in the study area and has a certain reference role for the extraction and analysis of late abnormal changes. We have found that the $CH_4$ concentration is greatly influenced by season, structure and underlying surface, and presents a typical spatial and temporal distribution in southwest China. Comparing our work with the GPS velocity field, the high value area of the background field corresponds to the area of high GPS velocity





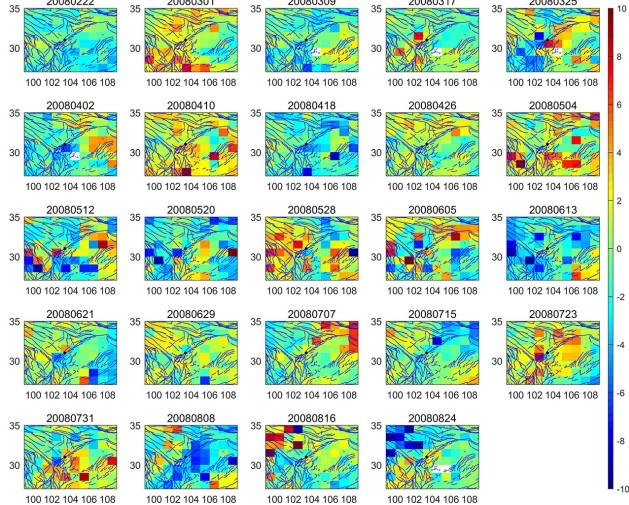

**Figure 10.** Distribution of the successive differential value at 300 hPa

(Wu et al., 2015). Long-term stress accumulation causes deep earth gases to be expelled along fissures or structural weaknesses (Wang et al., 2017).

Three indices have been used to see if anomalous $CH_4$ could be identified. The $CH_4$ spatial variations were most likely caused by geological processes and/or the action of crustal stress in the lithosphere resulting in the Wenchuan earthquake. A large amount of $CH_4$ was emitted from underground into the atmosphere along plate boundaries and tectonic belts from approximately $1.5$ months before to $3$ months after the earthquake, and the closer to the epicenter, the larger the amount of emitted gas. To further verify that the increase in $CH_4$ concentration is related to earthquakes, we compared and analyzed the variation of overall methane concentration and anomaly index in the study area for nonseismic years (Figures 11 and 12, there was no large earthquake during 2003 to 2007). Compared with 2003-2007 (Figure 11), the $CH_4$ concentration increased significantly in 2008 with an average increase of $5.12 * 10^{-8}$, compared with the average increase of $1.18 * 10^{-8}$ in the past five years (Figure 11). Especially between Mar 25th and June 21st. The results from the past five years showed that the concentration of $CH_4$ increased significantly due to seasonal influence from June 21st, showing a rapid increase in the slope of the curve (Table 1). Although there was a certain increase in 2008, the curve slope was relatively small (Table 1), which may be due to the large amount of $CH_4$ released by the aftershock earthquake. The increasing trend of $CH_4$ concentration caused by seasonal change weakened between Mar 25th and June 21st, 2008.

We also compare the Alice result with 2008 and 2007 (Figures 12 and 13). The distribution of the Alice index of 8-day $CH_4$ VMR at 300 hPa during 2007 is very different from 2008. In 2007, the index results were similar on the whole to stable features with no significant differences. The average value is almost between $-2$ to $2$ except Feb 7th, 2007. However, the average value for 2008 is over 2 after April 10th, 2008, one month before the earthquake. Although gaseous anomalies can be affected by climate change and weather patterns, human activity, vegetation coverage and many other factors beyond seismic activity, these





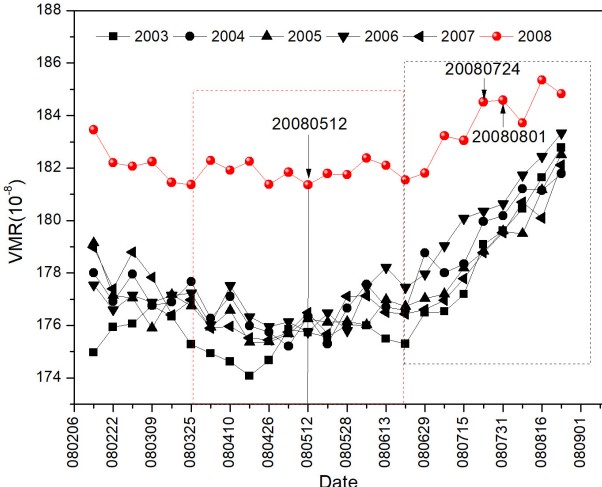

**Figure 11.** Comparison of the average $CH_4$ VMR in the study area from 2003 to 2008 (300 hPa)

**Table 1.** Linear fitting between June 21st and September 1st for different years

| Year | Linear equation | R-square |
|------|-----------------|----------|
| 2003 | $y = 0.9272x + 174.15$ | 0.9788 |
| 2004 | $y = 0.6044x + 176.53$ | 0.9051 |
| 2005 | $y = 0.6916x + 175.51$ | 0.9404 |
| 2006 | $y = 0.715x + 176.76$ | 0.9847 |
| 2007 | $y = 0.7058x + 175.25$ | 0.9484 |
| 2008 | $y = 0.4387x + 181.43$ | 0.8066 |

effects may be local and can be eliminated by calculation with Equations 1 to 5. Therefore, the identified anomalies of $CH_4$ VMR can be considered earthquake-related (Cui et al., 2017). $CH_4$ is an important greenhouse gas next to carbon dioxide, and previous work shows that the obvious anomalies of thermal infrared brightness temperature appeared on April 25th and were mainly distributed in the LMS fault zone and its southern region (Zhang et al., 2010). These results correspond well with our

5    work.

Compared with the nonseismic years from 2003 to 2007 (no earthquake >Ms 5.0), the study area before and after the Wenchuan earthquake showed a significantly increased $CH_4$ anomaly. It is well known that $CH_4$ is in a supercritical or critical state at high temperature and pressure 10-20 km below the surface, and its physical and chemical properties are relatively stable during periods of weak tectonic activity. However, with active geological activity, this state is also very easy to break.

10   For example, the sudden expansion of the volume of cracks or cavities caused by the change of tectonic stress field can easily change the internal fluid state and cause the supercritical fluid phase transition (Liu and Du, 2000). As soon as the phase transition process begins, fluid volume and fracture expansion are triggered, resulting in a large amount of fluid spilling over





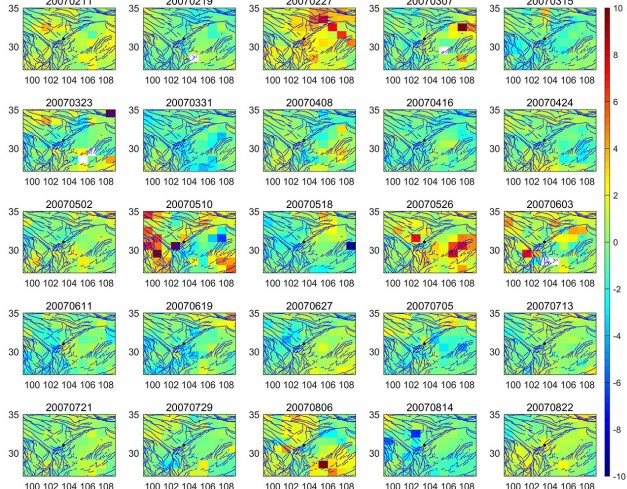

**Figure 12.** Alice index of 8-day $CH_4$ VMR at 300 hPa during 2007

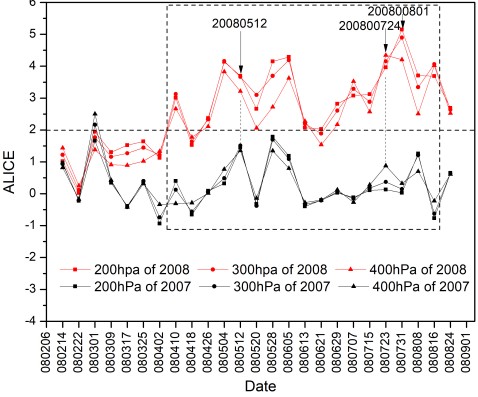

**Figure 13.** Comparison of the average Alice index from 2007 and 2008

the surface and spreading into the atmosphere, which is then observed by ground-based instruments or satellites (Martinelli
and Plescia, 2005; Italiano et al., 2008).

Before the Wenchuan earthquake, various GPS observation points and level observations showed accelerated activity of dif-
ferent degrees. The LMS Fault activity is characterized by a transition from tensile to compressive, and acceleration occlusion
5   occurs near the fault field. Accelerated uplift and extension of the crust appeared in the Bayan har block (Fang et al., 2009;
Jiao et al., 2008) indicating that when the main section of the LMS fault zone is still in the state of friction lock, the relatively
weak Bayan Har block shows the uplift and expansion of a large number of tensile cracks under long-term tectonic stress. It
is with this process that the underground fluid surges up and overflows the surface on a large scale, and then spreads and rises



to the upper troposphere, resulting in the high $CH_4$ concentration region on the western side of the LMS in Fig. 5-10 before the earthquake. Many NW fracture surfaces were found on the western side of the LMS after the Wenchuan earthquake. With the occurrence of such cracks, microcracks or preslip, supercritical $CH_4$ inside the crust can well up and overflow, resulting in increased atmospheric $CH_4$ concentration.

Figures 5 to 10 show that the $CH_4$ concentration in the inner area of the Sichuan Basin on the eastern side of the LMS Fault zone is at a low level before and after the earthquake, and occasionally increases temporarily. The occasional short rise may be due to $CH_4$ release and diffusion from the surrounding fault zone. Although there are a large number of oil and gas deposits in the sedimentary cover of the Sichuan Basin, the present tectonic activity is relatively weak, and it is difficult to produce such phenomena as rupture and expansion in the upper crust, so that large-scale release of $CH_4$ and other fluids will not occur.

The migration or release of underground fluids such as $CH_4$ is inevitably completed by other seismic processes in the pre-earthquake period, which is also an important way for energy conversion and accumulation from the deep part of the crust and mantle to the shallow part of the crust. Therefore, it can be inferred that the phase transition process of underground fluids such as $CH_4$ was mainly completed before the earthquake, so there was an abnormal increase before the earthquake. During the earthquake and coseismic activity, the abnormal increase of fluid concentration such as $CH_4$ may also occur, which is

mainly caused by local fluid release in the brittle layer of the upper crust (Wang et al., 2017). The range is mainly limited to the vicinity of the earthquake rupture zone, which is not on the same order of magnitude as the regional $CH_4$ release from the deep part. For example, the gradient shows that the values are positive before May 12th and become weak and even turn negative after May 12th ( Figures 8 and 9 ). The $CH_4$ in the upper layer may be deep gas released before the earthquake, and locally released after the earthquake decreases, so the concentration of the lower layer decreases and the gradient becomes negative.

The iteration of positive and negative values can also indicate geogenic emissions.

## 5    Conclusions

This study has examined the spatiotemporal variation in $CH_4$ in the mid-upper troposphere during the Wenchuan earthquake (May 12th, 2008) using AIRS retrieval data and discusses the mechanism of the methane anomaly. The $CH_4$ concentration increased significantly in 2008 with an average increase of $5.12*10^{-8}$, compared with $1.18*10^{-8}$ in the past five years. Three

indices were proposed and used. The Absolute Local Index of Change of the Environment (ALICE) was used for anomaly detection; the Vertical Concentration Gradient (Gradient) was proposed to study vertical variation; the Successive Differential Value (Diff) shows time variation. The three indices analyzed the spatial and temporal distribution of $CH_4$ before and after the earthquake from horizontal, vertical and time scales. Through this work, the following conclusions are obtained:

The background field of $CH_4$ is greatly influenced by season, structure and underlying surface and presents a typical spatial

and temporal distribution of methane in southwest China. Gas emanates from the earth's crust continuously, even without earthquakes.

The $CH_4$ concentrations in the upper and middle atmosphere before and after the Wenchuan earthquake have a certain temporal and spatial variation. The Alice and Diff indices could be used to identify the $CH_4$ concentration anomaly. The research



results indicate that the $CH_4$ concentration distribution before and after the earthquake breaks the distribution features of the background field. The $CH_4$ concentration distribution starts from both ends and gradually gathers around the epicenter. Moreover, a large anomalous area occurs centered at the epicenter eight days before the earthquake, and a trend of strengthening, weakening and strengthening appears over time.

The Gradient method can reflect the change in gas concentration in the vertical direction. The results show that the vertical direction obviously increases before the main earthquake, and the value is positive. The gradient value is negative during coseismic or postseismic events. It may be that the gas before the main earthquake mainly comes from deep fluid phase transitions, whereas the coseismic or postseismic gas may mainly come from the locally closed fluid in the brittle layer of the crust. However, due to the lack of any technical means to accurately identify the source and content of methane in the

atmosphere before the earthquake, an in-depth discussion has not been conducted, and further studies on this issue may be needed. The gradient index can reflect the characteristics of gas emission to some extent.

*Author contributions.*   Jing Cui designed this study, performed most of the data analysis and wrote the paper. Xuhui Shen contributed to the

discussions.

*Competing interests.*   The authors declare no competing interests.

*Acknowledgements.*   This research was supported by grants from the National Natural Science Foundation of China (Grant No. 41602223) and the Institute of Crustal Dynamics, China Earthquake Administration (Grant No. ZDJ2018-18). The authors would like to thank NASA for making available the AIRS dataset. The authors wish to thank the anonymous reviewers for their constructive comments that helped

improve the scholarly quality of the paper.





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
