# Peer review of "Analysis of Spatiotemporal variations in mid-upper tropospheric methane during the Wenchuan Ms8.0 earthquake by three indices"

_Natural Hazards and Earth System Sciences, 2018_

## Referee Comment (RC1) · Anonymous Referee #1 · 9 May 2019

**Review of the paper**

**Analysis of Spatiotemporal variations in mid-upper tropospheric methane during the Wenchuan Ms8.0 earthquake by three indices**

**By Jing Cuiand Xuhui Shen**

The paper faces very challenging topics related to the anomalous changes of CH4 atmospheric distribution (based on AIRS retrieval data) in possible relation with the occurrence of Wenchuan earthquake (12 May, 2008).

The paper is well written and the reviewer was delighted by the author's attempt to investigate CH4 dynamics in the temporal, vertical and horizontal dimensions.

However (as authors frequently complain along their manuscript) CH4 normal seasonal dynamics is apparently not considered in the computation of ALICE index and the concept of anomaly itself not completely developed in the choice of the

Reference fields used in the ALICE index computation should be separately computed for different season (e.g. monthly for the same month over 5 years) in order to take into account of CH4 seasonality avoiding that seasonal variability completely mask the ones possibly related to the considered EQ and to increase the S/N ratio by reducing standard deviation.

Even if Diff indices can partly compensate for variation of CH4 due to the seasonal cycle, their absolute values (without any comparison with a reference value and a normal variability like in an ALICE scheme) are not sufficient for appreciating the significance of reported variation as "anomalous"

The reviewer strongly suggests to publish this paper as soon as authors have improved it in the sense I suggested.

Minor comment. Quality of figures (dimensions and resolutions) should be improved. Acronyms should be in capital letters (e.g. Alice→ALICE). The quotation of Tramutoli et al, 2013 paper is more appropriate at paragraph 10 (after Tronin, 2006, Ouzounov et al. 2007) instead than at paragraph 30

---

## Referee Comment (RC2) · Anonymous Referee #2 · 9 May 2019

The manuscript investigates the spatiotemporal variation in methane in the mid-upper tropospheric before during and after the Wenchuan earthquake. On the whole, the paper is well presented and written, with efforts from the authors to present clearly the main ideas and concepts. They show that the methane concentration distribution before during and after the earthquake broke the distribution features of the background field.

A crucial question in this field of research refers to how can we link an individual precursor with a distinctive stage of the EQ preparation. In this direction, we focus on

the result presented by the authors that "a large anomalous area was centered at the epicenter area Âńeight daysÂż before the earthquake occurred".

The generation of such a seismic anomaly requires physical and chemical transformations which occur in a spatially extended preparation (activation) zone of an impending EQ.

Earthquakes exhibit in general complex correlations in time, space and magnitude. It is widely accepted that the observed EQ scaling laws indicate the existence of phenomena closely associated with the proximity of the system to a critical point [1]. Therefore, such a requirement is satisfied during the appearance of the "critical window", i.e., the epoch during which the short-range correlations have evolved to long-range ones in an extended area, where the "critical radius R" is given by the empirical relation logR  $\approx$  0,5M, where M is the EQ magnitude [2]. Notice, based on the recently introduced concept of the "natural time" by Varotsos and his colleagues [3] it has been shown that the foreshock seismic activity that occurs in the region around the epicentre of the upcoming significant shock Ańa few days up to one weekAż before the main shock occurrence, behaves as critical phenomenon.

Therefore, the hypothesis that the large anomaly in methane Âńeight daysÂż before the earthquake occurred corresponds to the critical point- window of the earthquake preparation process cannot be excluded. Accumulated experimental evidence supports the aforementioned hypothesis as follows:

The EQ preparatory process has various facets which reflect correspondingly different precursors. Importantly, precursors emerge during the same period, Âńa few days up

to one weekÂż before the main shock occurrence, while they behave as critical phenomena, as well. Characteristically, such as precursors are: (i) ULF magnetic field variations recorded by ground-based magnetic observatories before significant EQs, e.g., [4,5] (ii) MHz fracture induced MHz EM anomalies [6]. The generation of such a seismic anomaly also requires physical and chemical transformations which occur in a spatially extended preparation (activation) zone of an impending EQ.

Characteristic precursors are athe short-lived seismo-ionospheric EM precursors and EM anomalies rooted in preseismic LAI-coupling [7,8]. Pulinets et al. [7] have provided strong evidence for the occurrence of ionospheric precursors well before the main shock: ionospheric precursors within Âń5 daysÂż before the seismic shock were registered in 73% of the cases for EQs with a magnitude 5, and in 100% of the cases for EQs with a magnitude 6.

The aforementioned results seem to support the hypothesis that the observed anomaly in terms of spatiotemporal variation in methane is rooted in the stage of critical pointepoch of the earthquake preparation process.

[1] P. A. Varotsos, N. V. Sarlis and E. S. Skordas, Natural time analysis: Important changes of the order parameter of seismicity preceding the 2011 M9 Tohoku earthquake in Japan, EPL, 125 (2019) 69001, doi: 10.1209/0295-5075/125/69001

[2] Bowman, D., Quillon, G., Sammis, C., Sornette, A., Sornette, D., 1998. An observational test of the critical

earthquake concept. J. Geophys. Res. 103, 24359-24372.

[3] Varotsos, P., Sarlis, N., Skordas, E.S., 2011. Natural Time Analysis: The New View of Time. Springer, Berlin.

[4] Y. Contoyiannis, S.M. Potirakis, K. Eftaxias, M. Hayakawa, A. Schekotov, Intermittent criticality revealed in ULF magnetic fields prior to the 11 March 2011 Tohoku earthquake (Mw=9), Physica A 452, 19–28 (2016)

[5] S. M. Potirakis, Y. Contoyiannis, T. Asano, M. Hayakawa, Intermittency-induced criticality in the lower ionosphere prior to the 2016 Kumamoto earthquakes as embedded in the VLF propagation data observed at multiple stations, Tectonophysics 722, 422-431 (2018)

[6] K. Eftaxias, S. Potirakis and Y. Contoyiannis, Four-Stage Model of Earthquake Generation in Terms of fracture-Induced Electromagnetic Emissions: A Review, in

Complexity of Seismic Time Series, Measurement and Application, Edited by Tamaz Chelidze, Filippos Vallianatos and Luciano Telesca, Elsevier, Netherlands, 2019

[7] Pulinets, S., Legen'ka, A.D., Gaivoronskaya, T.V., Depuev, V. Kh, 2003. Main phenomenological features of

ionospheric precursors of strong earthquakes. J. Atmos. Sol. Terr. Phys. 65, 1337\_1347.

[8] Pulinets, S., Boyarchuk, K., 2004. Ionospheric Precursors of Earthquakes. Springer, Berlin.

---

## Author Comment (AC1) · 19 Jun 2019

Comment 1: However (as authors frequently complain along their manuscript) CH4 normal seasonal dynamics is apparently not considered in the computation of ALICE index and the concept of anomaly itself not completely developed in the choice of the Reference fields used in the ALICE index computation should be separately computed for different season (e.g. monthly for the same month over 5 years) in order to take into account of CH4 seasonality avoiding that seasonal variability completely mask the ones possibly related to the considered EQ and to increase the S/N ratio by reducing

standard deviation. Even if Diff indices can partly compensate for variation of CH4 due to the seasonal cycle, their absolute values (without any comparison with a reference value and a normal variability like in an ALICE scheme) are not sufficient for appreciating the significance of reported variation as "anomalous".

Reply: I accept the reviewer's advice. Maybe we did not present clearly. For the three index mentioned in the paper, we have consider the season and caculate the reference fields separately. We have re-written the part 2( Data and method), the Eqs and the parameters have been described again. For all the Eqs used in this paper, N was defined as 5 years, from 2003 to 2007; t is the time of the measurement acquisition ( t within a range where the range defines the homogeneous domain of the satellite imagery collected in the same time-slot of the day and period(month) of the year); l was defined as 3 layers (400 hPa, 300 hPa and 200 hPa).

For the Gradient index and Diff index, we have re-written and the result have been recalculated by the author's advice. All this two index have reducing standard deviation in the revised paper.

All the changes can been shown in the changed-marked-manuscript( as a supplement). The changes for this comment mainly in Part 2 ( Data and Method), and the changes marked with red color.

Minor comment1: Quality of figures (dimensions and resolutions) should be improved.

Reply: All the figures have been replaced. The resolutions have improved to 600dpi. For the Fig2-Fig10 and Fig12, the dimensions was changed from 5*5 to 5*4 and the label have been reset.

Minor comment2: Acronyms should be in capital letters (e.g. Alice to ALICE).

Reply: For the Fig5-Fig7 and Fig12-Fig13, we have change the Alice to ALICE and the manuscript also changed.

Minor comment3: The quotation of Tramutoli et al, 2013 paper is more appropriate at

paragraph 10 (after Tronin, 2006, Ouzounov et al. 2007) instead than at paragraph 30.

Reply: We accept the reviewer's advice. Tramutoli et al, 2013 paper have been quoted in Line 3,Page 2 of the changes-marked-manuscript with yellow color. This reference have been deleted in Line 10, Page 4 of the changes-marked-manuscript.

Please also note the supplement to this comment:
https://www.nat-hazards-earth-syst-sci-discuss.net/nhess-2018-342/nhess-2018-342-AC1-supplement.pdf

**Supplement:**

**Analysis of Spatiotemporal variations in mid-upper tropospheric methane during the Wenchuan Ms8.0 earthquake by three indices**

Jing Cui1 and Xuhui Shen1

1Key Laboratory of Crustal Dynamics, Institute of Crustal Dynamics, China Earthquake Administration, Beijing 100085, China

Correspondence: Jing Cui (jingcui\_86@yahoo.com)

**Abstract.** This research studied the spatiotemporal variation in methane in the mid-upper troposphere during the Wenchuan earthquake (12 May, 2008) using AIRS retrieval data and discussed the methane anomaly mechanism. Three indices were proposed and used for analysis. Our results show that the methane concentration increased significantly in 2008, with an average increase of  $5.12 \times 10^{-8}$ , compared to the average increase of  $1.18 \times 10^{-8}$  in the previous five years. The Alice and Diff indices

- 5 can be used to identify methane concentration anomalies. The two indices showed that the methane concentration distribution before and after the earthquake broke the distribution features of the background field. As the earthquake approached, areas of high methane concentration gradually converged towards the west side of the epicenter from both ends of the Longmenshan fault zone. Moreover, a large anomalous area was centered at the epicenter eight days before the earthquake occurred, and a trend of strengthening, weakening and strengthening appeared over time. The Gradient index showed that the vertical direction
- 10 obviously increased before the main earthquake, and the value was positive. The gradient value is negative during coseismic or postseismic events. The gradient index reflects the gas emission characteristics to some extent. We also determined that the methane release was connected with the deep crust-mantle stress state, as well as microfracture generation and expansion. However, due to the lack of any technical means to accurately identify the source and content of methane in the atmosphere before the earthquake, an in-depth discussion has not been conducted, and further studies on this issue may be needed.

15 Copyright statement. TEXT

**1 Introduction**

The great Wenchuan Ms8.0 earthquake, on May 12, 2008, occurred in the Longmenshan Fault zone in western Sichuan Province, China. Its epicenter was located at coordinates  $30.95^{\circ}N$ ,  $103.40^{\circ}E$ . The extent of the earthquake and aftershock affected areas in the northeast along the Longmen Shan fault, a thrust structure along the border of the Tibetan Plateau and the

20 western Sichuan Basin (Figure 1). This earthquake was one of the worst continental earthquake events to have struck China in recent decades, and it killed more than ten thousand people in several cities along the western Sichuan basin. A surface rupture more than 200 km long formed along the Yingxiu-Beichuan fault. The Guanxian-Jiangyou fault was also ruptured by

---

## Author Comment (AC2) · 19 Jun 2019

Advices: A crucial question in this field of research refers to how can we link an individual precursor with a distinctive stage of the EQ preparation. In this direction, we focus on the result presented by the authors that "a large anomalous area was centered at the epicenter area eight days before the earthquake occurred". ЁŹ The generation of such a seismic anomaly requires physical and chemical transformations which occur in a spatially extended preparation (activation) zone of an impending EQ. Earthquakes exhibit in general complex correlations in time, space and magnitude. It is widely accepted that the observed EQ scaling laws indicate the existence of phenomena closely associated with the proximity of the system to a critical point [1]. Therefore, such a requirement is satisfied during the appearance of the "critical window", i.e., the epoch during which the short-range correlations have evolved to long-range ones in an extended area, where the "critical radius R" is given by the empirical relation $logR \approx 0.5M$, where M is the EQ magnitude [2]. Notice, based on the recently introduced concept of the "natural time" by Varotsos and his colleagues [3] it has been shown that the foreshock seismic activity that occurs in the region around the epicentre of the upcoming significant shock a few days up to one week before the main shock occurrence, behaves as critical phenomenon. Therefore, the hypothesis that the large anomaly in methane eight days before the earthquake occurred corresponds to the critical point- window of the earthquake preparation process cannot be excluded. Accumulated experimental evidence supports the aforementioned hypothesis as follows: The EQ preparatory process has various facets which reflect correspondingly different precursors. Importantly, precursors emerge during the same period, A few days up to one week before the main shock occurrence, while they behave as critical phenomena, as well. Characteristically, such as precursors are: (i) ULF magnetic field variations recorded by ground-based magnetic observatories before significant EQs, e.g., [4,5] (ii) MHz fracture induced MHz EM anomalies [6]. The generation of such a seismic anomaly also requires physical and chemical transformations which occur in a spatially extended preparation (activation) zone of an impending EQ. Characteristic precursors are athe short-lived seismo-ionospheric EM precursors and EM anomalies rooted in preseismic LAI-coupling [7,8]. Pulinets et al. [7] have provided strong evidence for the occurrence of ionospheric precursors well before the main shock: ionospheric precursors within 5 days before the seismic shock were registered in 73% of the cases for EQs with a magnitude 5, and in 100% of the cases for EQs with a magnitude 6. The aforementioned results seem to support the hypothesis that the observed anomaly in terms of spatiotemporal variation in methane is rooted in the stage of critical pointepoch of the earthquake preparation process.

Reply: Thanks for the reviewer. This is a good advices, we have consider this, but do not know how to explain. We just compare our result with the non-earthquake year and try to explain this by the crust dynamics. We accept the reviewer's advice and add these advices in the Discussion part. The changes marked with red color in the Page 13-17 of the changed-marked-manuscript (shown as supplement). The reference also have been quoted with yellow color in the changed-marked-manuscript.

Please also note the supplement to this comment:
https://www.nat-hazards-earth-syst-sci-discuss.net/nhess-2018-342/nhess-2018-342-AC2-supplement.pdf